# Validation of a Length-Adjusted Abdominal Arterial Calcium Score Method for Contrast-Enhanced CT Scans

**DOI:** 10.3390/diagnostics13111934

**Published:** 2023-06-01

**Authors:** Raul Devia-Rodriguez, Maikel Derksen, Kristian de Groot, Issi R. Vedder, Clark J. Zeebregts, Reinoud P. H. Bokkers, Robert A. Pol, Jean-Paul P. M. de Vries, Richte C. L. Schuurmann

**Affiliations:** 1Department of Surgery, Division of Vascular Surgery, University Medical Center Groningen, University of Groningen, 9713 GZ Groningen, The Netherlands; 2Department of Radiology, Medical Imaging Center, University Medical Center Groningen, University of Groningen, 9700 RB Groningen, The Netherlands

**Keywords:** vascular diseases, atherosclerosis, calcified plaques, calcium score, four-dimensional computed tomography

## Abstract

Background: The Agatston score on noncontrast computed tomography (CT) scans is the gold standard for calcium load determination. However, contrast-enhanced CT is commonly used for patients with atherosclerotic cardiovascular diseases (ASCVDs), such as peripheral arterial occlusive disease (PAOD) and abdominal aortic aneurysm (AAA). Currently, there is no validated method to determine calcium load in the aorta and peripheral arteries with a contrast-enhanced CT. This study validated a length-adjusted calcium score (LACS) method for contrast-enhanced CT scans. Method: The LACS (calcium volume in mm^3^/arterial length in cm) in the abdominal aorta was calculated using four-phase liver CT scans of 30 patients treated between 2017 and 2021 at the University Medical Center Groningen (UMCG) with no aortic disease. Noncontrast CT scans were segmented with a 130 Hounsfield units (HU) threshold, and a patient-specific threshold was used for contrast-enhanced CTs. The LACS was calculated and compared from both segmentations. Secondly, the interobserver variability and the influence of slice thickness (0.75 mm vs. 2.0 mm) was determined. Results: There was a high correlation between the LACS from contrast-enhanced CT scans and the LACS of noncontrast CTs (*R*^2^ = 0.98). A correction factor of 1.9 was established to convert the LACS derived from contrast-enhanced CT to noncontrast CT scans. LACS interobserver agreement on contrast-enhanced CT was excellent (1.0, 95% confidence interval = 1.0–1.0). The 0.75 mm CT threshold was 541 (459–625) HU compared with 500 (419–568) HU on 2 mm CTs (*p* = 0.15). LACS calculated with both thresholds was not significantly different (*p* = 0.63). Conclusion: The LACS seems to be a robust method for scoring calcium load on contrast-enhanced CT scans in arterial segments with various lengths.

## 1. Introduction

Atherosclerosis is a progressive systemic disease and one of the leading causes of death [1,2]. High calcium load in the abdominal aorta, coronary arteries, and peripheral arteries has been associated with a higher risk of cardiovascular events and death [3,4].

Several methods have been described to quantify the arterial calcium load. The Agatston score is calculated by identifying individual calcium deposits and multiplying them by the volume of these deposits with a weighting factor determined by the maximum density of the deposit. The sum of all lesions results in the total calcium score, which classifies patients into four risk categories. An absolute Agatston score of 0, 1 to 100, 101 to 400, and >400 indicates very low risk, low risk, increased risk, and increased likelihood of future coronary events, respectively [5,6].

Most calcium score methods have been developed for noncontrast computed tomography (CT) scans and typically use a threshold of 130 Hounsfield units (HU) for segmenting the calcium. In patients with atherosclerotic cardiovascular diseases (ASCVDs) such as peripheral arterial occlusive disease (PAOD), abdominal aortic aneurysm (AAA), and carotid arterial stenosis, only contrast-enhanced CTs are commonly available. The standard threshold of 130 HU cannot be used in contrast scans because the blood contrast attenuation exceeds this threshold [7]. Therefore, a method to calculate calcium scores on contrast-enhanced CT is needed for patients with ASCVDs [8,9,10,11]. A dynamic patient-specific threshold was proposed by Raggi et al. to calculate the Agatston score on contrast-enhanced CT scans. This method adjusts the threshold values using three times the standard deviation (SD) above the mean contrast attenuation of a segmentation of the arterial blood volume [8,12]. Additionally, the length of the evaluated arterial segment varies between patients, and, therefore, the calcium score should be corrected for the length of the arterial segment.

Concretely, abdominal aortic calcium scores have helped identify patients’ risk of cardiovascular events and mortality [13,14]. Benjamens et al. classified patients prior to kidney transplantation in low (0–859), medium (860–5600), and high (>5600) aorto-iliac CaScore values (adapted Agatston score), demonstrating that patients in the medium and high score categories developed higher rates of cardiovascular events and all-cause mortality [14]. Similarly, O’Connor et al. found that asymptomatic patients with higher abdominal aorta Agatston scores presented an increased risk of cardiovascular events during a mean follow-up of 11.2 years, outperforming the classic Framingham cardiovascular risk score [13]. Therefore, implementing abdominal aortic calcium scores may improve patient risk stratification besides optimizing individual and preventive care.

The aim of this study was to validate a length-adjusted calcium score (LACS) to determine the calcium load on contrast-enhanced CT scans in arterial segments with varying lengths. The LACS determined from contrast-enhanced CT scans was compared with noncontrast CT scans with a 130 HU threshold (gold standard). The secondary aims were to determine interobserver variability and the influence of slice thickness on the established threshold and, consequently, the LACS.

## 2. Materials and Methods

A single-center, retrospective analysis of 4-phase liver CT scans obtained from patients between 2017 and 2021 at the University Medical Center Groningen (UMCG) was performed. The UMCG Institutional Ethical Review Board approved this study (study no. 202200343) and waived patient-informed consent for retrospective analysis. All study procedures were performed according to the Medical Research Involving Human Subjects Act and the Declaration of Helsinki. The study used 4-phase liver CT scans because they include both noncontrast and contrast-enhanced arterial phase series of the abdominal aorta. The LACS calculated from the noncontrast CT scans was considered the gold standard in this study.

### 2.1. Study Inclusion Criteria

From a large dataset of 4-phase liver CT scans, patients who met the following inclusion criteria were selected: (1) age > 55 years (because the prevalence of ASCVDs and incidence of arterial calcification increases considerably above this age [3]), (2) CT scans with a field of view from the celiac trunk to the aortic bifurcation, and (3) CT scan slice thickness reconstructions of 2.0 mm with 1.5 mm increments.

One exclusion criterion was considered in the study: CTs of patients with an endograft or stent in situ or other (metal) artifacts in the surrounding tissue of the abdominal aorta. Clinical outcomes were not considered in this study.

### 2.2. CT Protocol

Four-phase liver CT scans were obtained with the Somatom Force, Somatom definition Flash, Somatom definition AS, or Somatom definition Edge of Siemens Healthineers (Siemens Healthcare, Erlangen, Germany). Scans were performed using a spiral acquisition, with a pitch of 0.8 s and a collimation of 128 × 0.6 mm. Scan parameters were adjusted for patient body types and set in the range of 70 to 140 kVp, with 84% at 100 kVp, 36–551 mA, and a field of view of 296 to 500 mm. After a noncontrast scan, 100 mL of contrast medium (Iomeron 350, Bracco Imaging, Milan, Italy) was administered with a flow rate of 4.0 mL/s. The arterial phase was scanned with bolus timing, whereby the trigger was set at a threshold of 120 HU in the descending aorta at the top of the liver. Scans were reconstructed with a slice thickness of 0.75 mm and slice increments of 0.5 mm, and with a slice thickness of 2.0 mm and slice increments of 1.5 mm.

### 2.3. Calcium Score Analysis

#### 2.3.1. Calcium Segmentation

All included liver CT scans were processed using Aquarius iNtuition 4.4.13.P6 software (TeraRecon, Inc., San Mateo, CA, USA). On the contrast-enhanced CT scans, the mean contrast attenuation and +3 SD were determined on axial slices at the level of the celiac trunk and just above the aortic bifurcation. The highest value of these 2 segmentations was used as a threshold. The aortic wall was not segmented. Patient-specific thresholds were determined by adding 3 SDs to the mean contrast attenuation (Figure 1A) [12]. Additionally, the length of the abdominal aorta between the celiac trunk and the aortic bifurcation was measured using a center lumen line (Figure 1D). For noncontrast scans, the threshold was set to 130 HU [7]. Calcium was only selected if 4 adjacent voxels were segmented (as previously described by McCollough et al. [15]).

These thresholds were used to manually select calcified segments in all axial slices of the abdominal aorta to determine the total calcium volume (Figure 1B). The option “visible slice only” was used to avoid selection of calcium/bone outside the aortic wall (Figure 1C). The calcium was segmented on 2 mm sliced noncontrast scans, and on both 0.75 mm and 2 mm sliced contrast-enhanced CT scans.

The LACS was calculated by dividing the total calcium volume by the length of the arterial segment (volume score in mm^3^/length of artery in cm). Besides the LACS, the calcified lesion count was also determined in both scans. No specific software was used for LACS calculation other than Aquarius.

#### 2.3.2. Interobserver Agreement

The calcium segmentation measurements on the 2 mm contrast-enhanced CT scans were performed by 2 independent observers (M.D. and K.G.) for all patients. The LACSs determined by the first observer (M.D.) were compared with the second observer’s outcomes to determine interobserver agreement.

#### 2.3.3. Influence of CT Slice Thickness

The influence of slice thickness was evaluated by comparing the threshold, and the LACS was determined on reconstructions with 0.75 mm and 2 mm slice thickness of the contrast-enhanced CT scans. The 0.75 mm reconstruction was not available for 2 patients. The threshold was determined separately for both reconstructions.

### 2.4. Statistical Analysis

Statistical analyses were performed using IBM SPSS Statistics 25 software (IBM Corp, Armonk, NY, USA). Continuous variables were tested for normality, and not normally distributed variables were expressed using the median and the interquartile range (IQR), and normally distributed variables were described using the mean and SD. The Mann–Whitney *U*-signed rank test was used to compare the continuous nonnormally distributed variables, with *p* < 0.05 considered statistically significant.

Bland–Altman plots and the intraclass correlation coefficient (ICC) were used to compare the LACS derived from contrast-enhanced and noncontrast CT scans for interobserver agreement and to compare slice thicknesses. The ICC and 95% confidence interval (CI) were calculated based on a single rating and 2-way mixed effects model (<0.5 = poor reliability, 0.5–0.75 = moderate reliability, 0.75–0.9 = good reliability, and >0.9 = excellent reliability) [16].

Linear regression was used to determine the correction factor to convert the LACS obtained from contrast-enhanced CT scans to the LACS from noncontrast CT scans.

## 3. Results

From a database of 333 patients who underwent a four-phase liver CT scan at the UMCG between 2017 and 2021, 30 four-phase liver CT scans met the inclusion criteria and were analyzed in this study. Half of the included patients were men, and the median age was 62.5 years (IQR = 59–67 years).

### 3.1. Calcium Score Analysis

#### 3.1.1. Noncontrast vs. Contrast-Enhanced CT

The median patient-specific threshold obtained by the first observer from the arterial phase CT scans was 500 HU (IQR = 416–560 HU). The median length of the abdominal aorta was 13.1 cm (IQR = 12.5–13.9 cm). The median LACS calculated from the noncontrast CT scans was 224 mm^3^/cm (IQR = 82–438 mm^3^/cm) compared with 105 mm^3^/cm (IQR = 32–213 mm^3^/cm) on contrast-enhanced CT scans (Figure 2C). The difference was significant (*p* = 0.04)*,* yet the ICC showed good reliability (ICC = 0.90, 95% CI = 0.78–0.95).

The correction factor derived from the linear regression to convert the contrast to the noncontrast LACS was 1.9 (*R*^2^ = 0.98, Figure 2A). After correction, there was no significant difference between the corrected arterial LACS and noncontrast LACS (*p* = 0.78; Figure 2B), and there was an excellent reliability (ICC = 0.90, 95% CI = 0.80–0.96).

The median calcium lesions count from noncontrast scans was 114 (IQR = 61–165), compared with 81 (IQR = 32–127) lesions on the contrast-enhanced CT scan (*p* = 0.20; Figure 2D).

#### 3.1.2. Interobserver Agreement

The median patient-specific threshold determined by the second observer on contrast scans was 484 HU (IQR = 406–568 HU, Figure 3A). The second observer’s threshold was not significantly different from the first observer’s threshold (*p* = 0.76). The LACS calculated by the second observer was 108 mm^3^/cm (IQR = 30–256 mm^3^/cm, Figure 3B), which was not significantly different from the first observer *(p* = 0.79). The agreement between the LACS of both observers was excellent (ICC = 1.00, 95% CI = 1.00–1.00).

The median calcium lesions count by the second observer was 83 (IQR = 34–140, Figure 3D). The difference between the two observers was not significant (*p* = 0.72).

#### 3.1.3. Influence of Slice Thickness

The median threshold of the twenty-eight 0.75 mm reconstructions was 541 HU (IQR = 459–625 HU, Figure 4A), which was not significantly different from the threshold of the 2.0 mm reconstructions (median = 500 HU, IQR = 419–568 HU, *p* = 0.15). The LACS calculated on the 0.75 mm reconstructions was 97 mm^3^/cm (IQR = 39–215 mm^3^/cm, Figure 4B), which was not significantly different from the LACS of the 2.0 mm reconstructions (median = 111 mm^3^/cm, IQR = 43–234 mm^3^/cm, *p* = 0.63). The agreement of the LACSs was excellent (ICC = 0.93, 95% CI = 0.98–0.10).

The median calcium lesions count from the 0.75 mm reconstructions was 259 (IQR = 126–399, Figure 4D). The difference between the number of lesions segmented in the 2.0 mm and 0.75 mm scan reconstructions was significant (*p* < 0.01).

## 4. Discussion

This study showed that with a patient-specific threshold, the arterial calcium load can accurately be determined on contrast-enhanced CT scans using a correction factor of 1.9. After correction, there was no significant difference with the gold standard reference of noncontrast scans. Moreover, slice thickness (with a range of 0.75–2 mm) of the CT scan seemed to have minimal influence on the calculated LACS.

Patient-specific thresholds were previously proposed in the literature for contrast-enhanced coronary and abdominal aortic calcium scoring [8,9,10,11,17]. Mylonas et al. used a patient-specific threshold of two SDs above the mean contrast attenuation because the authors believed that the three SDs method (described by Raggi et al.) would exclude too many lower-attenuating calcifications. In this study, two-SD mean contrast attenuation thresholds were not assessed, but three SDs seems to correlate well with the gold standard.

Other patient-specific thresholds were proposed by Bischoff et al. and Buijs et al. for contrast-enhanced coronary and abdominal aortic calcium scoring, respectively [11,17]. Bischoff et al. proposed a patient-specific threshold of 150% of the mean contrast attenuation, which generally resulted in a higher threshold than the mean attenuation +3 SD, which might have resulted in underestimation of the calcium load.

Buijs et al. used four-phase liver scans to compare the volume score determined in contrast and noncontrast scans. Patient-specific thresholds were calculated using the global thresholding principle of digital images processing, which distinguishes calcium and contrast using a histogram. No correction factor was calculated, and, therefore, they concluded that volume scores determined on contrast-enhanced CT scans were not reliable enough for clinical use [11]. The mean patient-specific threshold in the Buijs et al. study was 230 ± 23 HU, which was much lower than the thresholds used in this study but resembled the 250 HU threshold used by Summer et al., who drew a similar conclusion and calculated a correction factor of 1.39 [9].

Other approaches have been proposed, such as converting the contrast-enhanced scans into noncontrast images by using strategies such as dual-layer spectral CT or using a deep-learning-based automated tool to recognize calcification content [9,18]. These methods showed good correlation with the Agatston score. Yet, they presented concerns, such as the increased radiation exposure of dual-source CT or the applicability in clinical practice.

The CT reconstructions with 0.75 mm slices had a higher noise ratio compared with 2.0 mm slice reconstructions. Therefore, a higher threshold and, consequently, less calcium segmentation was assumed on the 0.75 mm compared with the 2.0 mm scans. The effect on the segmented volume was not significant, but significantly fewer lesions were segmented. This indicates that small lesions may be overlooked on 0.75 mm reconstructions but that these small lesions do not contribute significantly to the total volume. These outcomes are in contrast with previous studies showing that coronary calcium volume scores determined on thinner slices were significantly higher compared with volume scores determined on thicker slices [19,20,21].

The interobserver analysis in this study showed excellent agreement and no significant deviation in the mean attenuation thresholds, the LACSs, and calcium lesion counts determined by both observers. Mylonas et al., Bijl et al., and Ghadri et al. also reported excellent interobserver agreement for Agatston scores determined with a patient-specific threshold on contrast-enhanced CT scans (ICC = 0.97) [8].

Factors not investigated in this study, but described in previous studies, are variabilities caused by different scanners, scanning and reconstruction parameters, and different software programs [5,22,23,24,25]. Previous studies reported good interscanner agreement for volume scores determined on multislice and dual-source CT scanners [5,22,23]. Four different CT scanners were used in this study. The influence of the scanners was visually inspected by the research team and was determined to be negligible. The limited data precluded us from performing a statistical test to determine potential differences between scanners.

The software that is used to segment the calcium volume has been found to significantly influence the calcium score. Siemens Syngo.via software overestimated phantom calcifications compared with GE SmartScore software and the ground truth in a previous study [22]. Moreover, different software programs were compared by Ajlan et al. (4DM Calcium score (INVIA, Ann Arbor, MI, USA), and Smart score (General Electric, Milwaukee, WI)) and Weininger et al. (Syngo Calcium Scoring (Siemens Healthcare, Erlangen, Germany), Aquarius (TeraRecon, San Mateo, CA, USA), and Vitrea (Vital Images, Minnetonka, MN, USA)), who found a high correlation between software outcomes [26,27].

### 4.1. Limitations

All approaches to determining the arterial calcium content using radiologic images represent an indirect measure, including the gold standard (noncontrast CT scans). This study used noncontrast CT scans to validate the LACS results of contrast-enhanced CT scans, which might have deviated from the real arterial calcium load. The LACS score validation did not evaluate the influence of CT scanners and analysis software, which might have influenced the calcium load calculations.

Furthermore, this study used liver CT scans to quantify calcium load in the abdominal aorta. The LACS, however, is intended to assess patient risk in ASCVDs. Thus, validation of the LACS must be tested on femoropopliteal arteries and correlated with outcomes of patients with ASCVDs to improve its clinical relevance.

### 4.2. Future Perspectives

The association of the LACS with clinical outcomes was not tested in this study and is important to know before the clinical application of the LACS. Therefore, in future studies, the LACS should be correlated with clinical outcomes of patients with vascular diseases to determine whether it can be used for risk stratification.

## 5. Conclusions

The LACS seems to be a robust method for scoring calcium load on contrast-enhanced CT scans in arterial segments with various lengths.

## Figures and Tables

**Figure 1 diagnostics-13-01934-f001:**
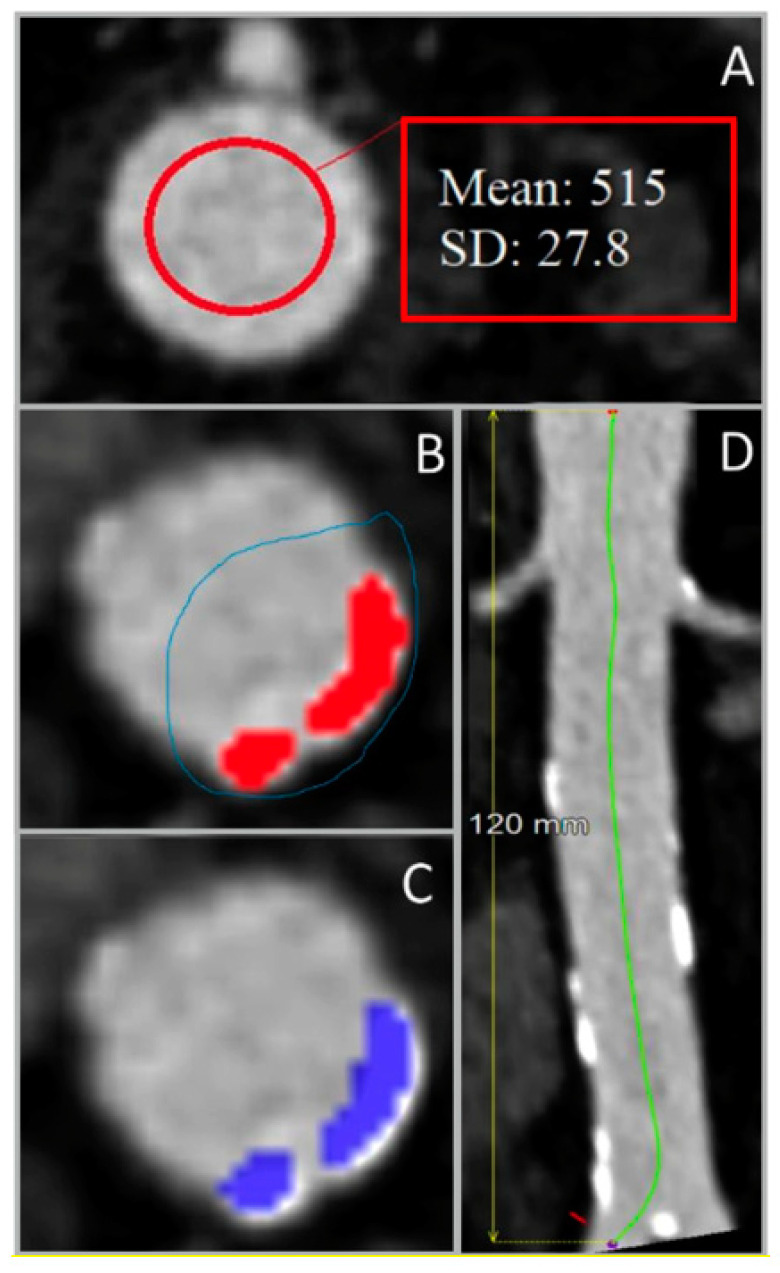
Steps for the determination of the length-adjusted arterial calcium score using abdominal aorta images from 4-phased liver contrast-enhanced CT scans. (**A**) Measurement of the contrast Hounsfield units (HU) in the upper region of interest, used for the determination of the patient-specific HU threshold. (**B**) Manual selection of the arterial calcified regions (colored in red) using the overlay. (**C**) Classification of calcified areas (colored in blue) by the preset HU threshold. (**D**) Measurement of the region of interest’s length using the center lumen line. SD: standard deviation.

**Figure 2 diagnostics-13-01934-f002:**
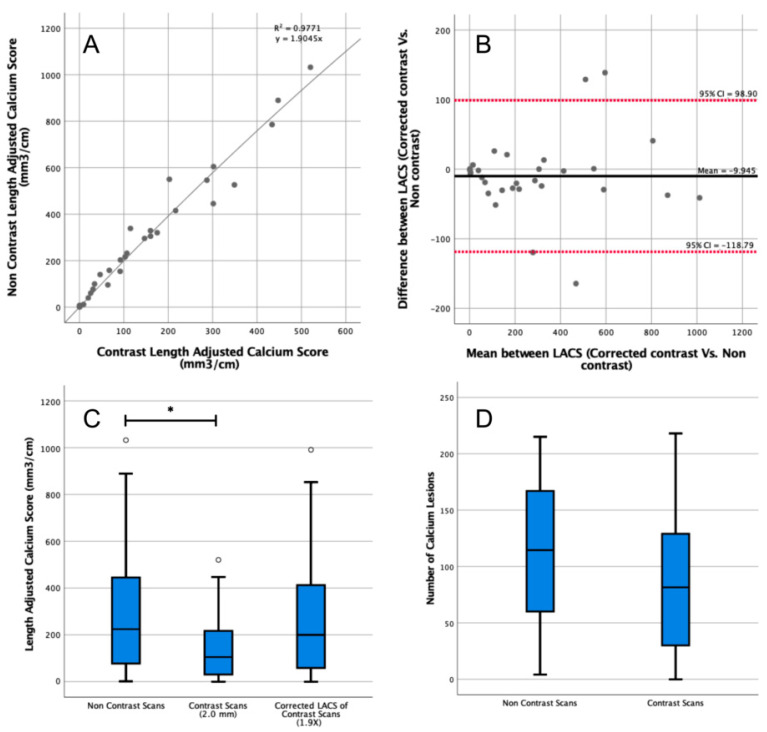
Length-adjusted arterial calcium score (LACS) determined in contrast-enhanced and noncontrast computed tomography (CT) scans. (**A**) Correction factor to convert from the LACS on contrast-enhanced to noncontrast CT of 1.9 was determined with linear regression. All data points are plotted as grey circles. (**B**) Comparison of the corrected LACS from contrast-enhanced scans (using the correction factor of 1.9) to LACS from noncontrast CT. CI: confidence interval. All data points are plotted as grey circles. (**C**) The LACS was determined in contrast-enhanced and noncontrast CT scans. Outlier data points are plotted as white circles. (**D**) Number of segmented calcium lesions in contrast-enhanced and noncontrast CT scans. * indicates *p* < 0.05.

**Figure 3 diagnostics-13-01934-f003:**
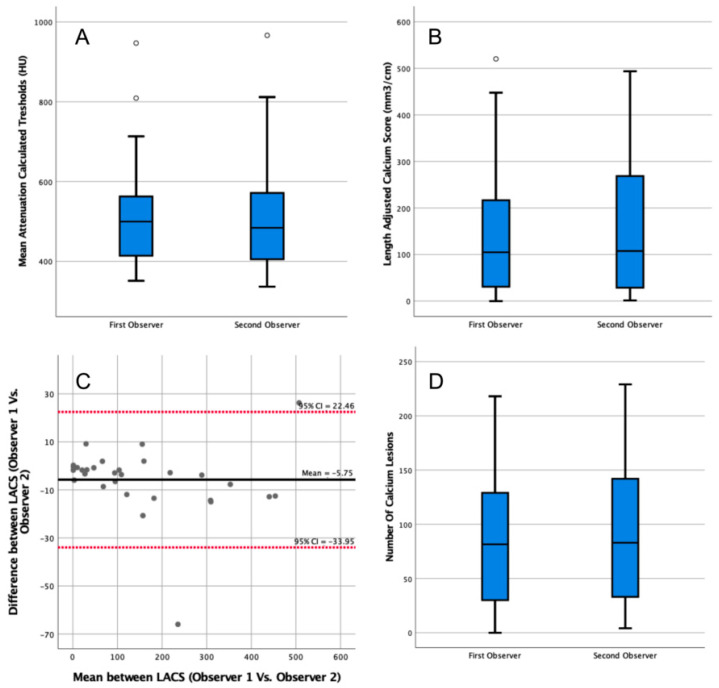
Analysis of the impact of interobserver variability in the length-adjusted arterial calcium score (LACS) outcomes determined in contrast-enhanced computed tomography (CT) scans. (**A**) Box plot comparing patient-specific thresholds calculated by two observers in contrast-enhanced CT scans. Outlier data points are plotted as white circles. (**B**) Box plot comparing the LACSs determined by two observers in contrast-enhanced CT scans. Outlier data points are plotted as white circles. (**C**) Bland–Altman plot comparing the LACSs determined in contrast-enhanced CT scans by two observers. CI: confidence interval. All data points are plotted as grey circles. (**D**) Box plot comparing the number of calcium lesions calculated by two observers in contrast-enhanced CT scans.

**Figure 4 diagnostics-13-01934-f004:**
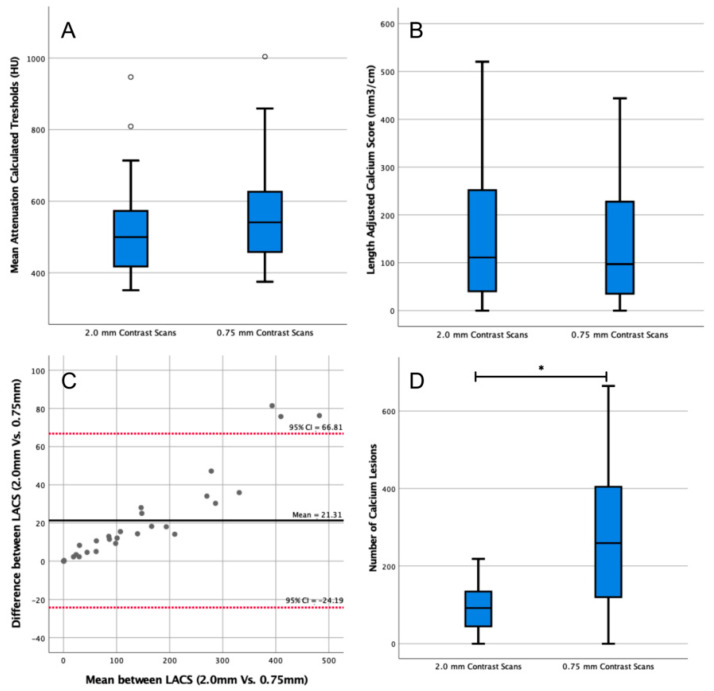
Analysis of the impact of slice thickness in length-adjusted arterial calcium score (LACS) outcomes. (**A**) Box plot comparing patient-specific thresholds calculated in 2.0 mm and 0.75 mm slice thickness contrast scans. Outlier data points are plotted as white circles. (**B**) Box plot comparing the LACS determined in 2.0 mm and 0.75 mm slice thickness contrast scans. (**C**) Bland–Altman plot comparing the LACS determined in contrast-enhanced computed tomography (CT) scans of two different thicknesses (2.0 mm vs. 0.75 mm). CI: confidence interval. All data points are plotted as grey circles. (**D**) Box plot comparing the number of calcium lesions calculated in 2.0 mm and 0.75 mm slice thickness contrast-enhanced CT scans. * indicates *p* < 0.05.

## Data Availability

The data presented in this study are available on request from the corresponding author. The data are not publicly available due to privacy and ethical reasons.

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
