# Peer review of "Validation of a Length-Adjusted Abdominal Arterial Calcium Score Method for Contrast-Enhanced CT Scans"

_diagnostics, 2023, doi:10.3390/diagnostics13111934_

Round 1

Reviewer 1 Report

This is interesting and sensible approach to quantifying non-coronary atherosclerotic  arterial calcification.  Missing is long term correlation with outcomes.

Author Response

Dear Reviewer,

Thank you for your insightful comments and feedback on the manuscript. Please find all the corrections in the rebuttal letter attached to this message.

Best,

Raul Devia Rodriguez

Reviewer 2 Report

The authors present the results of a study aimed at validating a length-adjusted calcium score (LACS) to determine the calcium load on contrast-enhanced CT scans in arterial segments with varying lengths.  Currently, data on the issue studied in this article are very limited. The results obtained by the authors are important for the development of valid patient-specific thresholds for measuring abdominal aortic calcium burden by contrast-enhanced CT. In the Discussion section, the authors of the study describe in detail the significance of the findings in the context of data already published on the topic. In my view, the methodology of the study supports the validity of these conclusions.  The list of references contains relevant publications and does not induce questions. As I have already noted, in my opinion, it would be convenient for the potential reader to see figures in the text as they are mentioned, rather than in a separate subsection. Therefore, I have only two minor points to consider:
1. The title of the manuscript should state that the abdominal aortic calcium load was studied
2. Figures in the manuscript should be incorporated into the text of the manuscript and follow their reference in the text.

Author Response

(The authors gave the same response as above.)

Reviewer 3 Report

The purpose of the study is to evaluate a length-adjusted calcium score (LACS) method for contrast-enhanced CT scans.

Abstract

-  In the background you wrote “contrast-enhanced CT is commonly used for patients with peripheral arterial disease” but this study would evaluate a length-adjusted calcium for the aorta, and I don’t understand the utility to introduce the peripheral arterial disease.

-  In the method I suggest: to explain how you calculate the LACS; to explain better the number of patients enrolled in the study; the period of the study.

-  Your conclusion are too definitive for a study with only 30 patients.

Introduction

-  I suggest evaluating better the literature about LACS reducing the literature about the PAOD. Why LACS could be important in the clinical practice?

Material and Method

- specify better the inclusion and exclusion criteria. Why have you decided to include patient > 55 years old. Eliminate the exclusion criteria from the results?

- I don’t understand if you have a specific software the calculate the LACS.

Results, Discussion and Conclusion

- The main important limitation is the utility of LACS in the clinical practice. I suggest explaining better this point.

- The conclusion are too definitive.

Images and tables are well explain.

Author Response

(The authors gave the same response as above.)

Round 2

Reviewer 3 Report

The author improved sufficiently their work.